# Racial differences in people living with HIV and Heart Failure: Insight from New York City health and hospitals HIV Heart Failure Cohort

Pawel Borkowski[1,2], Luca Biavati[1], Natalia Nazarenko[1], Matthew Parker[1], Amrin Kharawala[1,3], Coral Vargas-Pena[1,4], Shivang Bhakta[1,5], Ishmum Chowdhury[1], Joshua Bock[1], Vibhor Garg[1], Robert Faillace[1,6], Leonidas Palaiodimos[1,7], Yi-Yun Chen[1,8]*

1 Internal Medicine Department, NYC Health and Hospitals/Jacobi, Bronx, New York, 2 Division of Cardiology, University of Pittsburgh Medical Center, Harrisburg, Pennsylvania, United States of America, 3 Department of Cardiology, University of Nebraska Medical Center, Nebraska, 4 Department of medicine, NYU Grossman School of Medicine, New York, 5 Department of Critical Care, Montefiore Medical Center, Bronx, New York, 6 Cardiology Department, NYC Health and Hospitals/Jacobi, Bronx, New York, 7 Department of Medicine, Albert Einstein College of Medicine, Bronx, New York, 8 Division of Cardiology, Rhode Island Hospital/Brown University Health, Rhode Island

* crystalchen7883@gmail.com

## Abstract

### Background

Racial disparities, an imbalance between the treatment of racial groups, in healthcare significantly affect the prognosis and treatment outcomes for people living with HIV (PLHIV) and heart failure (HF). The complexity of racial disparities in health care is exacerbated when social determinants of health (SDoH). Utilizing the New York City Health and Hospitals HIV Heart Failure (NYC 4H) cohort, one of the largest public health providers in New York City, this study aims to describe the epidemiological characteristics, treatment, and mortality differences among various racial groups in patients living with HIV (PLHIV) and HF.

### Methods

This study utilized data from the mixed retrospective and prospective NYC 4H cohort, comprised of adult individuals with confirmed HIV and HF from inpatient or clinic visits between July 2017 and June 2022.
from eleven major New York City Health and Hospitals. Racial identification was reported by the patients. Social adversities (SA) were assessed through a psycho-social evaluation conducted by licensed clinical social workers (LCSWs) during the initial clinic or hospital encounter within the enrollment period. Each patient's home address was mapped to the area deprivation index (ADI) to obtain ADI ranking and further characterize socioeconomic disadvantage. We assessed the relationship between social adversities and overall mortality in each racial group using hazard ratios (HRs) derived from proportional hazard regression models.

**Data availability statement:** The data underlying this study contain sensitive patient information and are subject to institutional and IRB restrictions. The data are not publicly available due to confidentiality of patient records. Data access requests may be submitted to the NYC Health + Hospitals Institutional Review Board (IRB) at 718-579-5339, which serves as the non-author institutional point of contact for data access inquiries. Researchers may be required to submit a formal data use agreement and IRB approval for access.

**Funding:** The author(s) received no specific funding for this work.

**Competing interests:** The authors have declared that no competing interests exist.

## Results

In total, 1044 patients, including 631 Black/African American, 289 Hispanic/Latino, 57 non-Hispanic White, 17 Asian/Pacific Islander, and 50 of unknown or other racial backgrounds were analyzed in the study. An average follow-up time is 3.8 years. Significant racial difference in ischemic cardiomyopathy, with the highest occurrence found in the Black/African American group (51%) were noticed comparing to Asian/Pacific Islander (2.3%) and Other/Unknown groups (5.6%) (P < 0.001). The rate of coronary artery bypass grafting (CABG) among the African American population (1.9%) is notably lower (P < 0.001) compared to other racial groups. Based on state-wide ADI ranking, all patients disproportionately clustered in the most disadvantaged neighborhood areas, however patients identifying as Black/African American and Hispanic/Latino were more likely to reside in areas with higher ADI score. Asian/Pacific Islander populations have experienced fewer social adversities, with 76.5% reporting no encounters with social adversity, compared to 41% for Black/African American, 44.3% for Hispanic/Latino, 45.6% for non-Hispanic White, and 34% for Other/Unknown groups. Non-Hispanic White group exhibited a higher prevalence of LGBTQ individuals at 10.5%, a identity not commonly listed among the top challenge for other racial groups (P = 0.002). During the follow-up period, a total of 259 deaths were recorded. The mortality rate is lowest in Asian/Pacific Islanders (11.8%), while comparing to non-Hispanic Whites (33.3%) and unknown or other races (50%).

## Conclusions

Significant differences exist in comorbidities, disease management, and social conditions among HIV and heart failure patients across five racial groups. The findings suggest that within impoverished multiethnic communities, it is crucial to conduct comprehensive screenings for social adversities across all racial groups, as social disadvantage may manifest in various ways.

## Introduction

Despite the advent of modern treatments, human immunodeficiency virus (HIV) infection and heart failure (HF) continue to represent significant individual and public health challenges as chronic conditions [1–3]. Racial disparities in health outcomes among individuals living with both HIV and heart failure (HF) remain a critical but underexplored area of research [4,5]. A complex interplay of biological, socioeconomic, and healthcare access factors likely contributes to the observed differences in morbidity and mortality across racial and ethnic groups. Biologically, variations in genetic predisposition, inflammation pathways, and comorbid conditions may potentially influence disease progression and response to treatment in racially diverse populations [6,7]. It is known that HIV viral suppression is associated with significantly reduced inflammation, which contributes to an increased risk of age-related

conditions like cardiovascular disease [8,9]. Access to high-quality, culturally competent healthcare also plays a pivotal role in timely and chronic antiretroviral (ART) and HF treatment; racial and ethnic minorities are more likely to encounter structural barriers, including underinsurance, geographic limitations, and implicit bias within healthcare systems [10,11]. Understanding the multifactorial contributors to racial inequities in this dual-diagnosis population is essential for designing targeted interventions that promote health equity and improve clinical outcomes.

The complexity of racial disparities in health care is further exacerbated when social determinants of health (SDoH), such as socioeconomic status, lack of family support, education level, substance abuse, and living conditions, also act as modulators [12,13]. Therefore, a thorough analysis of how race and social challenges intersect is crucial for understanding their cumulative effect on health management and outcome across different racial groups. Yet, little research has examined racial differences among patients diagnosed with both HIV infection and heart failure (HF). Utilizing the New York City Health and Hospitals HIV Heart Failure (NYC 4H) cohort, one of the largest public health providers in New York City, this study aims to describe the epidemiological characteristics and treatment differences among various racial groups in patients with HIV and HF. Furthermore, we navigate how SDoH impact on racial disparity and survival.

## Methods

### Study population

This study utilized data from the NYC 4H cohort, consolidating records from eleven major New York City Health and Hospitals [14]. The dataset integrates a mix of retrospective baseline data collection and ongoing prospective follow-up details. The original NYC 4H cohort comprised adult individuals (Age ≥ 18) with diagnosis code of HIV and HF from inpatient or clinic visits between July 2017 and June 2022. After excluding patients without confirmed HIV infection and heart failure (HF) based on individual laboratory results and echocardiographic data, a total of 1,044 patients were included in the final cohort. Baseline demographic, medical, treatment, and social factors data were obtained through chart review at enrollment. The first follow-up period commenced from the first clinical encounter diagnosing HIV and HF to June 2022. A second round of detailed follow-up data was then gathered in subsequent hospital or clinic encounters from July 2022 through October 2023.

### Clinical variables

Participants self-reported their baseline demographic information, which was then documented in the electronic health record (EHR) [14]. Racial and ethnic classifications were based on self-reported data. Past medical histories were compiled from chart reviews. Medication information for both baseline and follow-up were verified using clinical encounter records in the EHR, with additional cross-checking against internal or external pharmacy records to resolve any duplications or discrepancies [15].

Heart failure with reduced ejection fraction (HFrEF) was defined as symptomatic heart failure with a left ventricular ejection fraction (LVEF) ≤40%; heart failure with mildly reduced ejection fraction (HFmrEF) as symptomatic heart failure with LVEF 41–49%; and heart failure with preserved ejection fraction (HFpEF) as symptomatic heart failure with LVEF ≥50%, consistent with the 2022 AHA/ACC/HFSA guideline definitions. Controlled HIV is defined by HIV viral load less than 200 copies/mL.

Functional status was determined based on the Activities of Daily Living (ADL). Patients were classified into three categories: completely independent (0 deficiencies in ADL), partially dependent (1–2 deficiencies in ADL), and completely dependent (more than 2 deficiencies in ADL), based on their individual ADL evaluations.

### Social adversities

Social adversities (SA) were assessed through a psychosocial evaluation conducted by LCSWs in New York State during the initial clinic or hospital encounter within the enrollment period. The comprehensive evaluation typically required approximately 45 minutes for a one-to-one personal interview. The spectrum of SA encompassed various factors, including criminal history, lack of insurance, immigration status, educational barriers, financial or job instability, lack of family

and community support, housing issues, substance abuse, mental illness, history of trauma, affiliation to Lesbian, Gay, Bisexual, Transgender and Queer (LGBTQ) community. The assessment of trauma history involved exploring experiences such as childhood abuse, being a crime victim, elder abuse, emotional abuse, human trafficking, intimate partner violence, neglect, physical abuse, sexual abuse, sexual assault, and other significant traumatic events.

Socioeconomic disadvantage was measured by using the area deprivation index (ADI), a standardized score based on census variables which combines measures of employment, income, housing, and education extracted from the American Community Survey [16]. The ADI scores for each zip code tabulation area (ZCTA) in New York State were linked to the ZCTA associated with each patient's permanent home address and analyzed as a continuous variable and by percentile ranking to compare the level of socioeconomic disadvantage experienced by cohort with the other ZCTAs in the state [17,18].

### Outcome

The Patient Outreach Department conducts annual follow-ups with patients who meet the criteria for loss of follow-up. Our primary outcome focused on the incidence of overall mortality during the follow-up period, which was identified first through the EHR. Between November 2023 and December 2023, all patients or their family members were contacted to verify the individual's survival status.

### Statistics analysis

Continuous variables and categorical variables were compared through independent t-test and chi-square test, respectively. We assessed the relationship between social adversities and overall mortality in each racial group, using hazard ratios (HRs) derived from proportional hazard regression models, adjusted for age, sex, baseline EF, controlled HIV, and comorbidities such as chronic obstructive pulmonary disease (COPD), end-stage renal disease (ESRD), cancer, hyperlipidemia, hypertension, diabetes mellitus, peripheral artery disease (PAD), pulmonary hypertension, and coronary artery disease (CAD), ADL and smoking status (S1 Table). Effect modification was tested via interaction terms (S2 Table). Sensitivity analysis was done among patient with controlled HIV and heart failure with reduced EF (HFrEF) (S3 Table). The Grambsch and Therneau test confirmed no violation of the proportional hazard assumption. All statistical tests were two-sided, with a significance threshold set at $P < 0.05$ to determine statistical significance.

Statistical analyses were performed using Stata (version 15.1; StataCorp) and R (version 4.3.3).

## Results

### Baseline characteristics

Table 1 provides an overview of the baseline characteristics for the NYC 4H cohort, broken down into five racial groups. The analysis included a total of 1044 patients, with distribution as follows: 631 Black/African American, 289 Hispanic/Latino, 57 non-Hispanic White, 17 Asian/Pacific Islander, and 50 of unknown or other racial backgrounds. The cohort was followed for an average duration of 3.8 years. Notably, the non-Hispanic White group showed a lower proportion of female participants (21%) compared to other racial groups. The Black/African American group exhibited a markedly higher prevalence of hypertension (80%), distinguishing it significantly from other races ($P < 0.001$). Asian group had the highest rate of never smoker (35.3%) compared to the rest of racial groups.

The non-Hispanic White group exhibited the lowest rate of controlled HIV, defined as an HIV viral load of less than 200 copies/ml [19], at 54.4% ($P = 0.02$). The highest median CD4 counts were observed in the non-Hispanic White (413 cells/mm^3), with the lowest recorded in the unknown/other racial category (255cells/mm^3). In terms of non-adherence to antiretroviral therapy (ART), the Hispanic/Latino population demonstrated the lowest non-compliance rate at 13.8%, followed by non-Hispanic White (14%) and Other/Unknown (14%).

**Table 1. Baseline characteristic by race in HIV HF patients.**

| | Black | Hispanic/Latino | Non-Hispanic white | Asian & pacific islander | Other/Unknown | P value |
|---|---|---|---|---|---|---|
| **Patient number** | 631 | 289 | 57 | 17 | 50 | |
| **Age, years** | 57.1 | 56.6 | 56.6 | 59.6 | 55.6 | 0.69 |
| **Sex, (%)** | | | | | | 0.02 |
| Female | 256 (40.6%) | 94 (32.5%) | 12 (21%) | 7 (41.2%) | 18 (36%) | |
| Male | 375 (59.4%) | 195 (67.5%) | 45 (79%) | 10 (58.8%) | 32 (64%) | |
| **Insurance, %)** | | | | | | 0.47 |
| Medicare | 246 (38%) | 98 (34%) | 31 (54.4%) | 6 (35.3%) | 17 (34%) | |
| Medicaid | 270 (42.8%) | 131 (45.5%) | 15 (26.3%) | 6 (35.3%) | 23 (46%) | |
| Commercial | 72 (11.4%) | 39 (13.5%) | 7 (12.3%) | 2 (11.8%) | 8 (16%) | |
| No insurance | 40 (6.3%) | 19 (6.6%) | 4 (7%) | 3 (17.6%) | 2 (4%) | |
| **Smoking, %)** | | | | | | 0.09 |
| Active smoker | 261 (41.4%) | 88 (30.5%) | 25 (43.9%) | 2 (11.8%) | 19 (38%) | |
| Former smoker | 205 (32.5%) | 118 (49.8%) | 19 (33.3%) | 9 (52.9%) | 21 (42%) | |
| Never smoker | 158 (25%) | 79 (27.3%) | 12 (21.1%) | 6 (35.3%) | 10 (25.4%) | |
| **Co-morbidity, %)** | | | | | | |
| Hypertension | 506 (80%) | 199 (68.9%) | 34 (60%) | 11 (64.7%) | 36 (72%) | <0.001** |
| Hyperlipidemia | 221 (35%) | 114 (39.5%) | 21 (36.8%) | 7 (41.2%) | 10 (20%) | 0.11 |
| Type 2 diabetes | 261 (41.5%) | 112 (38.9%) | 18 (31.6%) | 7 (41.2%) | 18 (36%) | 0.4 |
| COPD | 250 (39.6%) | 116 (40.1%) | 22 (38.6%) | 2 (11.8%) | 19 (38%) | 0.23 |
| CKD | 150 (23.8%) | 57 (19.7%) | 10 (17.5%) | 3 (17.7%) | 10 (20%) | 0.56 |
| **HIV** | | | | | | |
| median CD4 counts | 393 | 375 | 413 | 335 | 255 | |
| Controlled HIV, %) | 463 (73.4%) | 191 (66.1%) | 31 (54.4%) | 12 (70.6%) | 34 (68%) | 0.02* |
| On ART, %) | 573 (90.8%) | 265 (91.8%) | 50 (87.7%) | 15 (88.2%) | 43 (86%) | 0.67 |
| ART Non-compliance | 119 (18.9%) | 48 (13.8%) | 8 (14%) | 3 (17.7%) | 7 (14%) | 0.16 |
| **Heart failure** | | | | | | |
| median EF, %) | 46.5 | 45 | 40 | 35 | 50 | |
| HFrEF, %) | 322 (51%) | 147 (50.9%) | 32 (56.1%) | 12 (70.6%) | 23 (46%) | 0.49 |
| Ischemic cardiomyopathy | 288 (45.6%) | 135 (46.7%) | 36 (63.2%) | 9 (52.9%) | 26 (52%) | 0.128 |

Asian group demonstrated the highest percentage of heart failure with reduced ejection fraction (HFrEF) at 70.6% and the lowest median baseline ejection fraction (EF) at 35%. There were no significant racial difference in ischemic cardiomyopathy, with the highest occurrence found in the Non-Hispanic White (63.2%), followed by Asian/Pacific islander (52.9%) and Other/Unknown race (52%).

### HIV infection and heart failure treatment

Table 2 showed the variance in the medical treatment, represented by prescription of guideline-directed medical therapy (GDMT), and intervention across different racial groups. At both baseline and follow-up, there were no significant differences observed in the prescription rates of GDMT, aspirin, other than aspirin anti-platelet agents, and statins, among the five racial groups. Significant improvement of GDMT prescription were noticed in follow-up while comparing to the baseline across all five racial groups. The rate of coronary artery bypass grafting (CABG) among the African American population (1.9%) is notably lower (P < 0.001) compared to other racial groups, such as Hispanic/Latino (3.1%), non-Hispanic White (8.8%), Asian/Pacific Islander (23.5%). No statistically significant differences in the rates of pacemaker implantation

**Table 2. Heart failure treatment across race.**

| | Black | Hispanic/Latino | Non-Hispanic white | Asian/ pacific islander | Other/Unknown | P value |
|---|---|---|---|---|---|---|
| **Baseline Prescription** | | | | | | |
| Evidence-based BB[1] | 267 (42.3%) | 117 (40.5%) | 22 (38.6%) | 6 (35.3%) | 21 (42%) | 0.94 |
| ACEI/ARB/ARNI [2] | 284 (45%) | 133 (46%) | 29 (50.9%) | 6 (35.3%) | 20 (40%) | 0.73 |
| SGLT2 inhibitor [3] | 14 (2.2%) | 5 (1.7%) | 2 (3.5%) | 0 (0%) | 1 (2%) | 0.89 |
| MRA [4] | 43 (6.8%) | 14 (4.8%) | 6 (10.5%) | 0 (0%) | 3 (6%) | 0.38 |
| Aspirin | 264 (41.8%) | 112 (38.8%) | 22 (38.6%) | 6 (35.3%) | 15 (30%) | 0.51 |
| Other anti-platelets | 50 (7.9%) | 36 (12.5%) | 9 (15.8%) | 3 (17.7%) | 6 (12%) | 0.07 |
| Statin | 284 (45%) | 134 (46.4%) | 27 (47.4%) | 6 (35.3%) | 22 (44%) | 0.91 |
| One or more GDMT | 280 (60.2%) | 176 (60.9%) | 33 (57.9%) | 9 (52.9%) | 29 (58%) | 0.96 |
| Two or more GDMT | 190 (30.1%) | 81 (28%) | 19 (33.3%) | 3 (17.6%) | 13 (26%) | 0.69 |
| **Follow-up Prescription** | | | | | | |
| Evidence-based BB[1] | 370 (58.6%) | 170 (58.8%) | 34 (59.7%) | 7 (41.2%) | 31 (62%) | 0.66 |
| ACEI/ARB/ARNI [2] | 330 (52.3%) | 160 (55.4%) | 29 (50.9%) | 9 (52.9%) | 33 (66%) | 0.4 |
| SGLT2 inhibitor [3] | 71 (11.3%) | 34 (11.8%) | 7 (12.3%) | 1 (5.9%) | 6 (12%) | 0.96 |
| MRA [4] | 76 (12%) | 33 (11.4%) | 4 (7%) | 3 (17.7%) | 4 (8%) | 0.64 |
| Aspirin | 315 (49.9%) | 136 (47.1%) | 25 (43.9%) | 8 (47.1%) | 27 (54%) | 0.78 |
| Other anti-platelets | 74 (11.7%) | 37 (12.8%) | 4 (7%) | 2 (11.8%) | 6 (12%) | 0.82 |
| Statin | 403 (63.9%) | 176 (60.9%) | 39 (68.4%) | 9 (52.9%) | 27 (54%) | 0.42 |
| One or more GDMT | 474 (75%) | 222 (76.8%) | 41 (71.9%) | 13 (76.5%) | 40 (80%) | 0.87 |
| Two or more GDMT | 269 (42.6%) | 128 (44.3%) | 25 (43.9%) | 6 (35.3%) | 27 (54%) | 0.56 |
| **CABG** | 12 (1.9%) | 9 (3.1%) | 5 (8.8%) | 4 (23.5%) | 2 (4%) | <0.001 |
| **Pacemaker** | 19 (3%) | 2 (0.7%) | 0 (0%) | 1 (5.9%) | 0 (0%) | 0.07 |
| **ICD** | 27 (4.3%) | 14 (4.9%) | 3 (5.3%) | 0 (0%) | 1 (2%) | 0.78 |
| **CRT-D** | 2 (0.3%) | 3 (1%) | 0 (0%) | 0 (0%) | 0 (0%) | 0.59 |

1. Beta-blocker

2. angiotensin converting enzyme inhibitors (ACEI)/angiotensin receptor blockers (ARB)/ angiotensin receptor/neprilysin inhibitor (ARNI),

3. sodium-glucose cotransporter-2 (SGLT2) inhibitors

4. mineralocorticoid receptor antagonist (MRA).

(P = 0.07), implantable cardioverter-defibrillator (ICD) usage (P = 0.78), and cardiac resynchronization therapy with defibrillator (CRT-D) (P = 0.59) were noticed across these groups.

## Social determinants of health

Table 3 and S1 Fig shows the social adversities encountered by each racial group. Asian/Pacific Islander populations have experienced fewer SA, with 76.5% reporting no encounters with social adversity, compared to 41% for Black/African American, 44.3% for Hispanic/Latino, 45.6% for non-Hispanic White, and 34% for Other/Unknown groups.

Polysubstance abuse (PSA) emerged as the most common social adversity among the Black/African American, Hispanic/Latino, and Other/Unknown race groups, showing a statistically significant difference (P = 0.03). Mental illness and difficulties with insurance were more prevalent in the non-Hispanic White and Asian/Pacific Islander groups, respectively, though these findings were not statistically significant (P = 0.3 for mental illness; P = 0.4 for insurance difficulties). Common adversities for all racial groups were PSA, mental illness, and a lack of family support. Notably, the non-Hispanic White group exhibited a higher prevalence of LGBTQ individuals at 10.5%, a social adversity not listed among the top concerns for other racial groups, with this difference reaching statistical significance (P = 0.002).

**Table 3. Social adversities evaluation across race.**

| | Black | Hispanic/ Latino | Non-Hispanic white | Asian/ pacific islander | Other/ Unknown | P value |
|---|---|---|---|---|---|---|
| **Social adversities exposure** | | | | | | |
| 0 SA | 259 (41%) | 128 (44.3%) | 26 (45.6%) | 13 (76.5%) | 17 (34%) | 0.19 |
| 1-2 SA | 269 (42.6%) | 112 (38.8%) | 22 (38.6%) | 3 (17.7%) | 23 (46%) | |
| >2 SA | 103 (16.3%) | 49 (17%) | 9 (15.8%) | 1 (5.9%) | 10 (20%) | |
| **Specific SA** | | | | | | |
| PSA | 214 (33.9%) | 77 (26.6%) | 13 (22.8%) | 2 (11.8%) | 19 (38%) | 0.03 |
| Mental illness | 130 (20.6%) | 61 (21.1%) | 16 (28.1%) | 1 (5.9%) | 8 (16%) | 0.30 |
| Housing difficulty | 71 (11.3%) | 36 (12.5%) | 8 (14%) | 1 (5.9%) | 5 (10%) | 0.87 |
| Food insecurity | 19 (3%) | 19 (3.5%) | 1 (1.8%) | 0 (0.00%) | 2 (4%) | 0.88 |
| Job insecurity | 35 (5.6%) | 20 (6.9%) | 0 (0%) | 0 (0%) | 6 (12%) | 0.07 |
| Financial insecurity | 39 (6.2%) | 21 (7.3%) | 2 (3.5%) | 1 (5.9%) | 4 (8%) | 0.84 |
| Family support | 84 (13.3%) | 43 (14.9%) | 11 (19.3%) | 2 (11.8%) | 11 (22%) | 0.38 |
| No insurance | 40 (6.3%) | 19 (6.6%) | 4 (7%) | 3 (17.7%) | 2 (4%) | 0.40 |
| Transportation | 44 (7%) | 24 (8.3%) | 0 (0%) | 0 (0%) | 3 (6%) | 0.16 |
| Education difficulty | 9 (1.4%) | 9 (3.1%) | 0 (0%) | 0 (0%) | 0 (0%) | 0.21 |
| Undocumented | 5 (0.8%) | 4 (1.4%) | 0 (0%) | 0 (0%) | 1 (2%) | 0.73 |
| Trauma history | 28 (4.4%) | 9 (3.1%) | 4 (7%) | 0 (0%) | 3 (6%) | 0.53 |
| Safety concern | 8 (1.3%) | 2 (0.7%) | 3 (5.3%) | 0 (0%) | 1 (2%) | 0.09 |
| Criminal history | 9 (1.4%) | 1 (0.3%) | 0 (0%) | 0 (0%) | 1 (2%) | 0.49 |
| LGBTQ | 19 (3%) | 4 (1.4%) | 6 (10.5%) | 0(0%) | 0 (0%) | 0.002 |

From the socioeconomic perspective, patients identifying as Black/African American and Hispanic/Latino resided in neighborhoods with the highest ADI scores, representing least resourced, when compared to other racial groups (**Fig 1**). However, when compared to all New York State ZIP Code Tabulation Area (ZCTA) -defined ADI rankings, all racial groups were disproportionately clustered in the more disadvantaged (higher ADI) ranks.

## Overall mortality across race

During the follow-up period, a total of 259 deaths were recorded. A variation in mortality rates was noted among different racial groups, with the lowest observed in Asian/Pacific Islanders (11.8%) and the highest in non-Hispanic Whites (33.3%) and individuals of unknown or other races (50%). **Fig 2** illustrates the risk of death for each racial group, as determined by a proportional hazard regression model. The analysis revealed that compared to the Asian population, individuals of other/ unknown races had a 4.6 times higher risk of death during the follow-up period (Hazard Ratio [HR] 4.57, 95% Confidence Interval [CI] [1.06, 19.8], P = 0.04).

## Discussion

Racial and ethnic minorities in the United States continue to bear a disproportionate burden of both HIV and HF [20,21,22]. To our knowledge, this is the first multi-center longitudinal study to examine differences in biological factors, SDoH, and medical implementation among people living with HIV (PLHIV) and HF across five distinct racial and ethnic groups. This study is uniquely positioned in that it draws from the diverse patient population served by the largest safety-net healthcare systems in New York State—systems that primarily care for individuals often excluded from traditional research. As such, our findings provide critical insights into the intersection of race, chronic illness, and healthcare disparities among underrepresented communities.

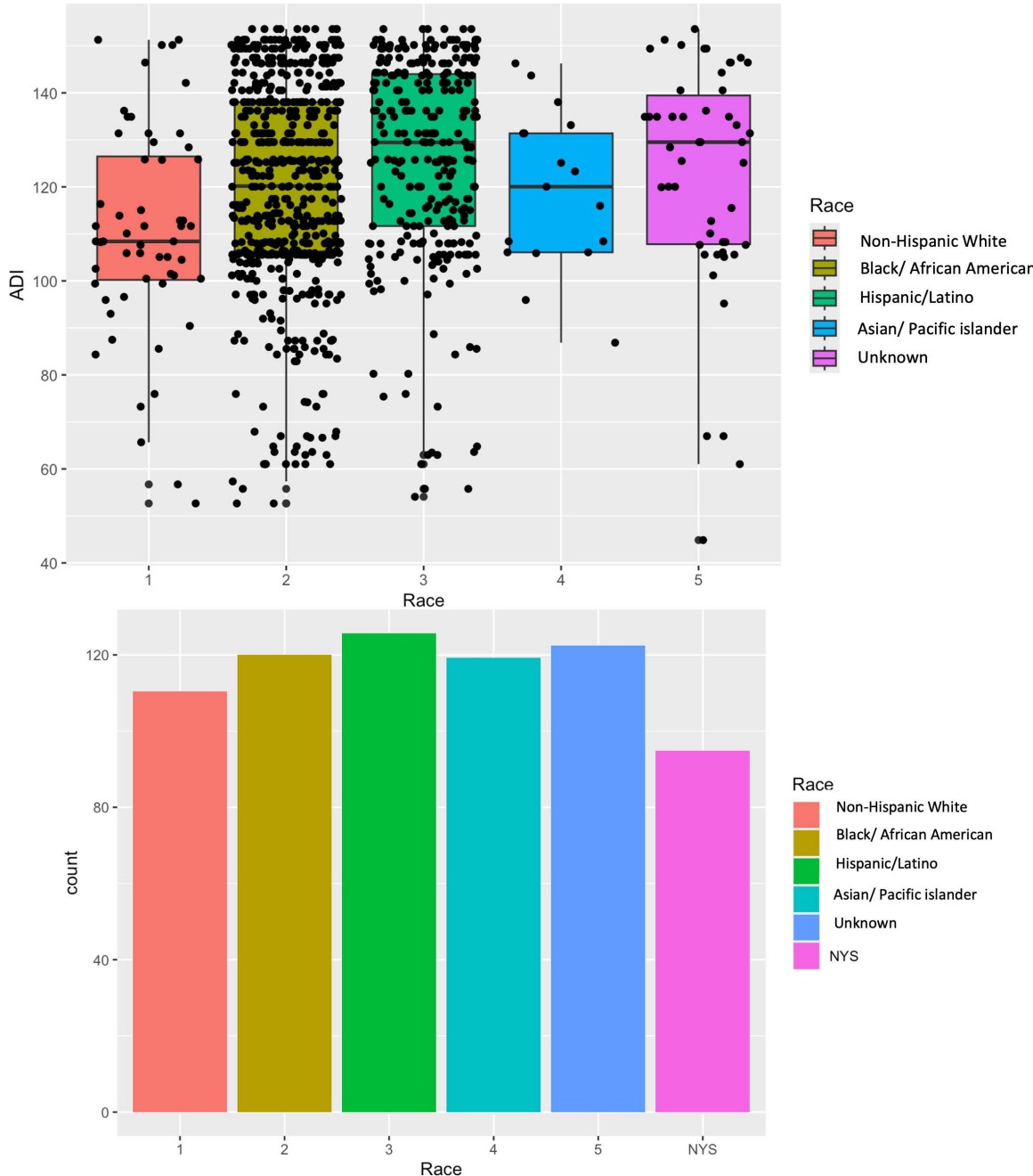

**Fig 1. Area Deprivation index across racial groups.** Caption: Patients identifying as Black/African American and Hispanic/Latino resided in neighborhoods with the highest ADI scores. However, when compared to all New York State ZIP Code Tabulation Area (ZCTA) -defined ADI rankings, all racial groups were disproportionately clustered in the more disadvantaged (higher ADI) ranks.

Our study demonstrates that the differences in outcomes observed across racial groups are multifactorial, reflecting the combined influence of biological, socioeconomic, and healthcare access–related factors.

Similar to prior studies, our study observed a higher prevalence of hypertension and ischemic cardiomyopathy among Black/African American individuals, demonstrating incidence and underlying disease may be partly contribute to the

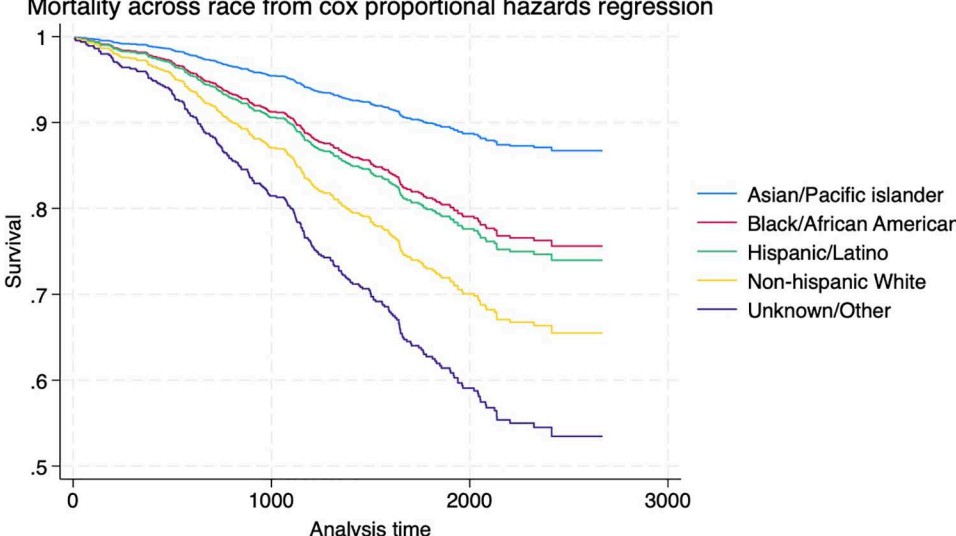

**Fig 2. Death Across Race By Proportional Hazard Regression Model.** Caption: The hazard ratio for mortality is 4.57 times higher in unknown/other groups, in comparison to Asian/Pacific islander.

difference. Furthermore, it is notable that coronary artery bypass grafting (CABG) rates were significantly lower in this population. Studies have demonstrated that Black/African American individuals utilize revascularization procedures less frequently compared to other racial groups, contributing to this differences [23,24]. Popescu et al. have similarly highlighted disparities between White and Black individuals in the quality of hospital care for acute myocardial infarction (AMI) and CABG surgery [25,26]. The implementation gap may partly explain the worse outcome in certain racial groups.

The proportion of individuals identifying as other/unknown race in our cohort is notably low (4.7%). The unexpectedly high burden of social adversity in individuals identifying as "other/unknown" race is consistent with literature highlighting the invisibility and marginalization of racially ambiguous or non-disclosed individuals [27,28]. The decision to withhold racial information or self-identify could serve as a potential entry point for SDoH within this vulnerable population. Studies have illuminated that individuals may refrain from disclosing their race due to discomfort or perceived threat stemming from past experiences of discrimination or marginalization [29,30]. This finding aligns with our observation that individuals of unknown/other racial backgrounds bear a heavier burden of social adversities, potentially motivating them to conceal their race to evade further scrutiny or targeted societal responses. This group's high mortality rate may reflect elevated allostatic load from chronic stress exposure, underutilization of health services, or unmeasured barriers to care such as stigma and trauma-related avoidance. Furthermore, the youngest age and highest mortality was observed in the unknown/other racial groups, indicating disparity in premature mortality among these specific groups and warrant further investigation [31].

Our study population demonstrated lower adherence to ART, which may reflect a combination of structural and individual barriers. These could include unstable housing, limited access to healthcare, mental health comorbidities, substance use, and stigma—all of which are more prevalent among individuals served by safety-net health systems. The Interestingly, despite receiving similar treatment regimens, non-Hispanic Whites in our study displayed the poorest control of HIV. This stands in contrast to the general epidemiology finding, where CDC data shows viral suppression rates usually are highest among this group [32]. Our investigators hypothesize that non-Hispanic Whites with higher socioeconomic status (SES) may be more likely to avoid public healthcare systems, potentially introducing bias into the perceived health outcomes of non-Hispanic whites who do utilize such services. Our analysis with patient's ZCTA-defined ADI showed that

patients enrolled in the study were more likely to reside in neighborhoods with greater disadvantage. Although the present study was not powered to assess the association between race/ethnicity and neighborhood deprivation because of measures gathered over large geographic areas (zip code-level data), we were able to link each patient to their socioeconomic surroundings and compare them with state-wide rankings. Our findings suggest that, among non-Hispanic White individuals who utilize public safety-net hospitals, a significant proportion experiences greater socioeconomic disadvantage than the national/state average. Furthermore, we also notice that non-Hispanic whites also exhibit higher and distinct social challenges, particularly regarding trauma history and LGBTQ status, in comparison to other racial groups. This finding suggests that social adversities extend beyond income alone, potentially impacting non-Hispanic Whites, who have historically been perceived as having higher socioeconomic status in other dimensions.

These results challenge the assumption that racial identity alone predicts health advantage and emphasize the role of intersecting adversities. Individuals across different racial group demonstrate their unique challenge and it underscores the importance of conducting comprehensive screenings for social adversities across all racial groups. These findings also highlight the necessity for further research to elucidate the specific factors and potential intervention in the social adversities, including socioeconomic, systemic, and possibly healthcare-related biases contributing to poorer health outcomes.

## Limitations

First, our study is limited by small sample size for certain racial groups, notably Asian/Pacific Islanders, which may not provide sufficient statistical power to identify significant associations. Additionally, the patient cohort was drawn exclusively from New York City, a region that, while demographically diverse, presents unique patient characteristics and medical settings not necessarily representative of those found in suburban or rural areas across the country, or within the private healthcare system. Furthermore, our cohort originates from New York City's largest safety-net hospital system, which is committed to providing healthcare services to the most vulnerable population, regardless of socioeconomic status, insurance coverage, or immigration status. This commitment ensures a unique study population that may significantly differ from those typically served by private healthcare systems, creating the selection bias and limited generalization. Third, racial data were derived from self-reported information, and we did not have an independent method to verify its accuracy. Lastly, despite extensive covariate adjustment, residual confounding is an inherent limitation of observational cohort studies and results should be interpreted with caution. However, our sensitivity analyses yielded consistent findings, supporting the robustness of the results.

## Conclusion

Significant differences exist in comorbidities, disease management, and social conditions among PLHIV and HF patients across five racial groups. The findings suggest that within impoverished multiethnic communities, it is crucial to conduct comprehensive screenings for social adversities across all racial groups in order to provide effective healthcare, as social disadvantage may manifest in various ways.

## Supporting information

**S1 Table. Race and Overall Mortality through Cox regression hazard model. Model adjusted for age, sex, baseline EF, controlled HIV, and comorbidities such as chronic obstructive pulmonary disease (COPD), end-stage renal disease (ESRD), cancer, hyperlipidemia, hypertension, diabetes mellitus, peripheral artery disease (PAD), pulmonary hypertension, and coronary artery disease (CAD), ADL and smoking status.**
(DOCX)

**S2 Table. Potential effect modifier with race to mortality.**
(DOCX)

**S3 Table. Sensitivity analysis within heart failure with reduced EF patient. Model adjusted for age, sex, baseline EF, controlled HIV, and comorbidities such as chronic obstructive pulmonary disease (COPD), end-stage renal disease (ESRD), cancer, hyperlipidemia, hypertension, diabetes mellitus, peripheral artery disease (PAD), pulmonary hypertension, and coronary artery disease (CAD), ADL and smoking status.**
(DOCX)

**S1 Fig. Most prevalent social adversities across racial groups. Percentage of reported needs by demographic category, highlighting the prevalence of Mental Health, PSA, and Family-related concerns across diverse populations.**
(PNG)

**Acknowledgments** The authors express their sincere gratitude to the staff and participants of the NYC 4H cohort for their invaluable contributions.

## Author contributions

**Conceptualization:** Pawel Borkowski, Joshua Bock, Vibhor Garg, Robert Faillace, Yi-Yun Chen.

**Data curation:** Pawel Borkowski, Luca Biavati, Natalia Nazarenko, Matthew Parker, Ishmum Chowdhury, Joshua Bock, Vibhor Garg.

**Formal analysis:** Pawel Borkowski, Luca Biavati, Vibhor Garg.

**Investigation:** Pawel Borkowski, Natalia Nazarenko, Coral Vargas-Pena, Shivang Bhakta, Ishmum Chowdhury, Yi-Yun Chen.

**Methodology:** Luca Biavati, Matthew Parker, Coral Vargas-Pena, Shivang Bhakta, Ishmum Chowdhury, Yi-Yun Chen.

**Supervision:** Robert Faillace, Leonidas Palaiodimos, Yi-Yun Chen.

**Visualization:** Pawel Borkowski.

**Writing – original draft:** Pawel Borkowski, Natalia Nazarenko, Amrin Kharawala, Yi-Yun Chen.

**Writing – review & editing:** Pawel Borkowski, Yi-Yun Chen.

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
