## [Decision Letter · Decision Letter 0]

16 Sep 2025

Dear Dr. Chen,

Thank you for submitting your manuscript to PLOS ONE. After careful consideration, we feel that it has merit but does not fully meet PLOS ONE’s publication criteria as it currently stands. Therefore, we invite you to submit a revised version of the manuscript that addresses the points raised during the review process.

**ACADEMIC EDITOR:**

The following changes are required for acceptance of your revised manuscript:

Use the term persons or patients with HIV instead of HIV-positive patients.

Write Health and Hospitals instead of Health + Hospitals.

Correct the sentence “Yet little can be found to address the racial differences in patient has both diagnosis HIV infection and HF'. in lines 55-56.

Place figures and captions right after the paragraph in which they are first cited and write the title in bold letters.

Explain the reasons for low HIV treatment adherence and rate of viral suppression across different racial/ethnic categories.

Consider adjusting for confounding variables or if not done, acknowledge this as study limitation.

The comments provided by Reviewer 3 are appropriate suggestions and recommendations.

This Journal Editor feels that your manuscript addresses a very important topic as persons living with HIV are living longer and experiencing a higher incidence of cardiovascular diseases, fueled by ongoing inflammation, even in the setting of consistent viral suppression. Moreover, people of lower socio-economic status are facing increased risk of cardiovascular comorbidity regardless of HIV status.

We look forward to receiving your revised manuscript.

Kind regards,

Vladimir Berthaud

Academic Editor

PLOS ONE

Journal Requirements:

2. We note you have included a table to which you do not refer in the text of your manuscript. Please ensure that you refer to Table 3 in your text; if accepted, production will need this reference to link the reader to the Table.

3. We note that there is identifying data in the Supporting Information file <file name>. Due to the inclusion of these potentially identifying data, we have removed this file from your file inventory. Prior to sharing human research participant data, authors should consult with an ethics committee to ensure data are shared in accordance with participant consent and all applicable local laws.

-Location data

Additional guidance on preparing raw data for publication can be found in our Data Policy (https://journals.plos.org/plosone/s/data-availability#loc-human-research-participant-data-and-other-sensitive-data) and in the following article: http://www.bmj.com/content/340/bmj.c181.long .

4. Please include captions for your Supporting Information files at the end of your manuscript, and update any in-text citations to match accordingly. Please see our Supporting Information guidelines for more information: http://journals.plos.org/plosone/s/supporting-information .

Additional Editor Comments:

Reviewer #1: Minor Revision: LINE 131: Brief background of why CD4 cells are important in HIV patients.

Fig. 1 : Replace 'prevalence' with 'prevalent'.

Do the socio-economic conditions affect the diet? Diet might have direct and indirect effect on heart disease.

Reviewer #2: Minor Revision

This manuscript presents a technically sound and well-executed study examining racial disparities in HIV-positive patients with heart failure using data from the NYC 4H cohort. The research is timely, relevant, and contributes meaningfully to the understanding of how social adversity intersects with clinical outcomes in marginalized populations.

1. Technical Soundness and Data Support

The study design, a mixed retrospective and prospective cohort, is appropriate for the research question. The sample size (n=1,044) is adequate, and the inclusion of five racial groups allows for meaningful stratified analysis. The use of validated tools such as the Area Deprivation Index (ADI) and psychosocial evaluations conducted by licensed social workers adds rigor to the assessment of social adversity.

The statistical analyses are appropriately chosen and executed. Proportional hazards regression models are used to assess mortality risk, with adjustments for key confounders. The manuscript confirms that proportional hazard assumptions were tested and not violated. Chi-square and t-tests are applied correctly to compare categorical and continuous variables, respectively. Hazard ratios and confidence intervals are clearly reported, and the findings are supported by well-organized tables and figures.

The conclusions are appropriately drawn from the data. The attenuation of mortality differences after adjusting for social adversity is a compelling finding that reinforces the manuscript’s central thesis.

2. Statistical Rigor

The statistical methodology is robust. The multivariable models are well-constructed, and the stratification by race and social adversity adds depth to the analysis. The authors have transparently acknowledged limitations related to small subgroup sizes and geographic specificity, which do not undermine the core findings.

3. Data Availability

The authors have confirmed that all relevant data are included within the manuscript and its Supporting Information files. This complies with PLOS ONE’s data policy and ensures transparency and reproducibility.

4. Language and Presentation

The manuscript is written in standard English and is generally intelligible. The structure is logical, and the tone is professional. However, a few grammatical and syntactical issues should be addressed:

• Revise awkward phrasing such as “Yet little can be found to address the racial differences in patient has both diagnosis HIV infection and HF.”

A light editorial review would enhance clarity and polish.

5. Figures and Tables Formatting

While figures and tables are cited appropriately in the Results and Discussion sections, the manuscript does not fully comply with PLOS ONE’s formatting guidelines:

• Figure captions should appear directly after the paragraph in which they are first cited. Currently, they are listed separately.

• Figure titles should be in bold type to distinguish them clearly.

• No tables are embedded within figure captions, which is correct and appreciated.

Addressing these formatting issues will improve readability and ensure compliance with journal standards.

6. Ethical Compliance and Publication Integrity

The study received IRB approval (Study ID 23-12-663719(HHC)), and all data were anonymized. The ethics statement is complete and appropriate. The authors have declared no competing interests and no funding sources, and there is no indication of dual publication.

Overall Recommendation: This manuscript meets the criteria for publication in PLOS ONE. With minor editorial and formatting revisions, it will make a valuable contribution to the literature on health disparities in HIV and heart failure populations.

Reviewer #3: Major revision

The manuscript addresses an important and timely topic, exploring racial differences in HIV-positive patients with heart failure. Overall, the study has potential to make a meaningful contribution, but several areas could be strengthened. First, the rationale for focusing on racial differences should be more clearly articulated in the introduction, with a discussion of potential biological, socioeconomic, and healthcare access factors that could contribute to observed disparities. The methodology section would benefit from additional clarity regarding patient selection, inclusion and exclusion criteria, and how race was categorized and verified. Details on how heart failure was defined and classified, including ejection fraction categories and relevant biomarkers, should be included to allow reproducibility. Statistical analyses need to be more explicitly described, including any adjustments for confounding variables such as age, sex, comorbidities, and antiretroviral therapy use; consideration of interaction terms or stratified analyses might also enhance interpretation. The results section would benefit from more granular reporting, particularly regarding subgroup differences and effect sizes, rather than focusing solely on statistical significance. Discussion of potential limitations is limited; the authors should address issues such as residual confounding, single-center design, and potential selection bias, and how these factors might influence generalizability. Additionally, the discussion could more thoroughly integrate the findings with existing literature, highlighting similarities, differences, and potential mechanistic explanations. Finally, attention to clarity in tables, figures, and text—ensuring consistency in terminology and proper labeling—would improve readability. Overall, the manuscript addresses a clinically relevant topic, but these areas of improvement would enhance rigor, clarity, and interpretability.

Reviewers' comments:

Reviewer's Responses to Questions

**Comments to the Author**

1. Is the manuscript technically sound, and do the data support the conclusions?

Reviewer #1: Partly

Reviewer #2: Yes

Reviewer #3: Yes

2. Has the statistical analysis been performed appropriately and rigorously?

Reviewer #1: N/A

Reviewer #2: Yes

Reviewer #3: Yes

3. Have the authors made all data underlying the findings in their manuscript fully available?

Reviewer #1: Yes

Reviewer #2: Yes

Reviewer #3: Yes

4. Is the manuscript presented in an intelligible fashion and written in standard English?

Reviewer #1: Yes

Reviewer #2: Yes

Reviewer #3: Yes

Reviewer #1: LINE 131: Brief background of why CD4 cells are important in HIV patients.

Fig. 1 : Replace 'prevalence' with 'prevalent'.

Do the socio-economic conditions affect the diet? Diet might have direct and indirect effect on heart disease.

Reviewer #2: This manuscript presents a technically sound and well-executed study examining racial disparities in HIV-positive patients with heart failure using data from the NYC 4H cohort. The research is timely, relevant, and contributes meaningfully to the understanding of how social adversity intersects with clinical outcomes in marginalized populations.

1. Technical Soundness and Data Support

The study design, a mixed retrospective and prospective cohort, is appropriate for the research question. The sample size (n=1,044) is adequate, and the inclusion of five racial groups allows for meaningful stratified analysis. The use of validated tools such as the Area Deprivation Index (ADI) and psychosocial evaluations conducted by licensed social workers adds rigor to the assessment of social adversity.

The statistical analyses are appropriately chosen and executed. Proportional hazards regression models are used to assess mortality risk, with adjustments for key confounders. The manuscript confirms that proportional hazard assumptions were tested and not violated. Chi-square and t-tests are applied correctly to compare categorical and continuous variables, respectively. Hazard ratios and confidence intervals are clearly reported, and the findings are supported by well-organized tables and figures.

The conclusions are appropriately drawn from the data. The attenuation of mortality differences after adjusting for social adversity is a compelling finding that reinforces the manuscript’s central thesis.

2. Statistical Rigor

The statistical methodology is robust. The multivariable models are well-constructed, and the stratification by race and social adversity adds depth to the analysis. The authors have transparently acknowledged limitations related to small subgroup sizes and geographic specificity, which do not undermine the core findings.

3. Data Availability

The authors have confirmed that all relevant data are included within the manuscript and its Supporting Information files. This complies with PLOS ONE’s data policy and ensures transparency and reproducibility.

4. Language and Presentation

The manuscript is written in standard English and is generally intelligible. The structure is logical, and the tone is professional. However, a few grammatical and syntactical issues should be addressed:

• Revise awkward phrasing such as “Yet, little can be found to address the racial differences in patient has both diagnosis HIV infection and HF.”

A light editorial review would enhance clarity and polish.

5. Figures and Tables Formatting

While figures and tables are cited appropriately in the Results and Discussion sections, the manuscript does not fully comply with PLOS ONE’s formatting guidelines:

• Figure captions should appear directly after the paragraph in which they are first cited. Currently, they are listed separately.

• Figure titles should be in bold type to distinguish them clearly.

• No tables are embedded within figure captions, which is correct and appreciated.

Addressing these formatting issues will improve readability and ensure compliance with journal standards.

6. Ethical Compliance and Publication Integrity

The study received IRB approval (Study ID 23-12-663719(HHC)), and all data were anonymized. The ethics statement is complete and appropriate. The authors have declared no competing interests and no funding sources, and there is no indication of dual publication.

Overall Recommendation: This manuscript meets the criteria for publication in PLOS ONE. With minor editorial and formatting revisions, it will make a valuable contribution to the literature on health disparities in HIV and heart failure populations.

Reviewer #3: The manuscript addresses an important and timely topic, exploring racial differences in HIV-positive patients with heart failure. Overall, the study has potential to make a meaningful contribution, but several areas could be strengthened. First, the rationale for focusing on racial differences should be more clearly articulated in the introduction, with a discussion of potential biological, socioeconomic, and healthcare access factors that could contribute to observed disparities. The methodology section would benefit from additional clarity regarding patient selection, inclusion and exclusion criteria, and how race was categorized and verified. Details on how heart failure was defined and classified, including ejection fraction categories and relevant biomarkers, should be included to allow reproducibility. Statistical analyses need to be more explicitly described, including any adjustments for confounding variables such as age, sex, comorbidities, and antiretroviral therapy use; consideration of interaction terms or stratified analyses might also enhance interpretation. The results section would benefit from more granular reporting, particularly regarding subgroup differences and effect sizes, rather than focusing solely on statistical significance. Discussion of potential limitations is limited; the authors should address issues such as residual confounding, single-center design, and potential selection bias, and how these factors might influence generalizability. Additionally, the discussion could more thoroughly integrate the findings with existing literature, highlighting similarities, differences, and potential mechanistic explanations. Finally, attention to clarity in tables, figures, and text—ensuring consistency in terminology and proper labeling—would improve readability. Overall, the manuscript addresses a clinically relevant topic, but these areas of improvement would enhance rigor, clarity, and interpretability.

**Do you want your identity to be public for this peer review?** For information about this choice, including consent withdrawal, please see our Privacy Policy

Reviewer #1: No

Reviewer #2: **Yes:** Edgar Muchinta

Reviewer #3: **Yes:** Chukwuka Elendu

---

## [Author Response · Author response to Decision Letter 1]

7 Nov 2025

ACADEMIC EDITOR:

The following changes are required for acceptance of your revised manuscript:

Use the term persons or patients with HIV instead of HIV-positive patients.

Author reply:

All of the term is changed to people living with HIV (PLHIV).

Write Health and Hospitals instead of Health + Hospitals.

Author reply:

All of the terms are changed to Health and Hospitals.

Correct the sentence “Yet little can be found to address the racial differences in patient has both diagnosis HIV infection and HF'. in lines 55-56.

Author reply:

The sentence is now corrected to “Yet, little research has examined racial differences among patients diagnosed with both HIV infection and heart failure (HF).”

Place figures and captions right after the paragraph in which they are first cited and write the title in bold letters.

Author reply:

All figures/captions/table are all placed right after the paragraph in which they were first cited.

Explain the reasons for low HIV treatment adherence and rate of viral suppression across different racial/ethnic categories.

Author reply:

We appreciate editors’ suggestion. A small paragraph were added to the discussion part to explain the lower adherence.

“Our study population demonstrated lower adherence to ART, which may reflect a combination of structural and individual barriers. These could include unstable housing, limited access to healthcare, mental health comorbidities, substance use, and stigma—all of which are more prevalent among individuals served by safety-net health systems. ”

Consider adjusting for confounding variables or if not done, acknowledge this as study limitation.

Author reply:

The proportional hazard model was adjusted with age, sex, baseline EF, controlled HIV, and comorbidities such as chronic obstructive pulmonary disease (COPD), end-stage renal disease (ESRD), cancer, hyperlipidemia, hypertension, diabetes mellitus, peripheral artery disease (PAD), pulmonary hypertension, and coronary artery disease (CAD), ADL and smoking status. It was mentioned in the results section and now moved up to the method section.

Reviewer #1: Minor Revision:

LINE 131: Brief background of why CD4 cells are important in HIV patients.

Author reply:

We include the introduction of how viral suppression is important for PLHIV.

“It is known that HIV viral suppression is associated with significantly reduced inflammation, which contributes to an increased risk of age-related conditions like cardiovascular disease. Access to high-quality, culturally competent healthcare also plays a pivotal role in timely and chronic antiretroviral (ART) and HF treatment; racial and ethnic minorities are more likely to encounter structural barriers, including underinsurance, geographic limitations, and implicit bias within healthcare systems.”

Fig. 1 : Replace 'prevalence' with 'prevalent'.

Author reply: Figure 1 is now replaced by different figure. But we change the title to prevalent in the supplementary information.

Do the socio-economic conditions affect the diet? Diet might have direct and indirect effects on heart disease.

Author reply: We appreciate and agree on the thought. Yes, we believe that SES affect the diet. However, the cohort did not build with the dietary information. The information will be gathered in the next data collection cycles. We include this in the last sentences in discussion. “These findings also highlight the necessity for further research to elucidate the specific factors, including dietary, socioeconomic, systemic, and possibly healthcare-related biases contributing to poorer health outcomes.”

Reviewer #2: Minor Revision

This manuscript presents a technically sound and well-executed study examining racial disparities in HIV-positive patients with heart failure using data from the NYC 4H cohort. The research is timely, relevant, and contributes meaningfully to the understanding of how social adversity intersects with clinical outcomes in marginalized populations.

1. Technical Soundness and Data Support

The study design, a mixed retrospective and prospective cohort, is appropriate for the research question. The sample size (n=1,044) is adequate, and the inclusion of five racial groups allows for meaningful stratified analysis. The use of validated tools such as the Area Deprivation Index (ADI) and psychosocial evaluations conducted by licensed social workers adds rigor to the assessment of social adversity.

The statistical analyses are appropriately chosen and executed. Proportional hazards regression models are used to assess mortality risk, with adjustments for key confounders. The manuscript confirms that proportional hazard assumptions were tested and not violated. Chi-square and t-tests are applied correctly to compare categorical and continuous variables, respectively. Hazard ratios and confidence intervals are clearly reported, and the findings are supported by well-organized tables and figures.

The conclusions are appropriately drawn from the data. The attenuation of mortality differences after adjusting for social adversity is a compelling finding that reinforces the manuscript’s central thesis.

2. Statistical Rigor

The statistical methodology is robust. The multivariable models are well-constructed, and the stratification by race and social adversity adds depth to the analysis. The authors have transparently acknowledged limitations related to small subgroup sizes and geographic specificity, which do not undermine the core findings.

3. Data Availability

The authors have confirmed that all relevant data are included within the manuscript and its Supporting Information files. This complies with PLOS ONE’s data policy and ensures transparency and reproducibility.

4. Language and Presentation

The manuscript is written in standard English and is generally intelligible. The structure is logical, and the tone is professional. However, a few grammatical and syntactical issues should be addressed:

• Revise awkward phrasing such as “Yet little can be found to address the racial differences in patient has both diagnosis HIV infection and HF.”

Author reply: The authors thank reviewers for the grammar correction. The sentences is now re-written as “Yet, little research has examined racial differences among patients diagnosed with both HIV infection and heart failure (HF).”

5. Figures and Tables Formatting

While figures and tables are cited appropriately in the Results and Discussion sections, the manuscript does not fully comply with PLOS ONE’s formatting guidelines:

• Figure captions should appear directly after the paragraph in which they are first cited. Currently, they are listed separately.

• Figure titles should be in bold type to distinguish them clearly.

• No tables are embedded within figure captions, which is correct and appreciated.

Addressing these formatting issues will improve readability and ensure compliance with journal standards.

Author reply: We thank reviewer for the suggestion. The article is revised and should be compatible with the PLOS One format.

6. Ethical Compliance and Publication Integrity

The study received IRB approval (Study ID 23-12-663719(HHC)), and all data were anonymized. The ethics statement is complete and appropriate. The authors have declared no competing interests and no funding sources, and there is no indication of dual publication.

Overall Recommendation: This manuscript meets the criteria for publication in PLOS ONE. With minor editorial and formatting revisions, it will make a valuable contribution to the literature on health disparities in HIV and heart failure populations.

Reviewer #3: Major revision

The manuscript addresses an important and timely topic, exploring racial differences in HIV-positive patients with heart failure. Overall, the study has potential to make a meaningful contribution, but several areas could be strengthened.

First, the rationale for focusing on racial differences should be more clearly articulated in the introduction, with a discussion of potential biological, socioeconomic, and healthcare access factors that could contribute to observed disparities.

Author reply:

We thank reviewer for the recommendation. We extensively edit the introduction to make it more focus on the racial disparity in HIV HF patients. Here is the paragraph in the introduction section:

“Racial disparities in health outcomes among individuals living with both HIV and heart failure (HF) remain a critical but underexplored area of research. A complex interplay of biological, socioeconomic, and healthcare access factors likely contributes to the observed differences in morbidity and mortality across racial and ethnic groups. Biologically, variations in genetic predisposition, inflammation pathways, and comorbid conditions such as hypertension and diabetes may potentially influence disease progression and response to treatment in racially diverse populations. Access to high-quality, culturally competent healthcare also plays a pivotal role; racial and ethnic minorities are more likely to encounter structural barriers, including underinsurance, geographic limitations, and implicit bias within healthcare systems. These disparities may result in delayed diagnosis, suboptimal treatment, and poorer outcomes. Understanding the multifactorial contributors to racial inequities in this dual-diagnosis population is essential for designing targeted interventions that promote health equity and improve clinical outcomes.”

The methodology section would benefit from additional clarity regarding patient selection, inclusion and exclusion criteria, and how race was categorized and verified.

Author’s reply:

1) We appreciate reviewer’s suggestions. We had included sentences to clarify the selection process “The original NYC 4H cohort comprised adult individuals (Age ≥18) with confirmed diagnosis code of HIV and HF from inpatient or clinic visits between July 2017 and June 2022. After excluding patients without confirmed HIV infection and heart failure (HF) based on individual laboratory results and echocardiographic data, a total of 1,044 patients were included in the final cohort.”

2) Racial data were derived from self-reported information, and we did not have an independent method to verify its accuracy. We have addressed this limitation explicitly in both the Methods and Limitations sections of the manuscript.

Details on how heart failure was defined and classified, including ejection fraction categories and relevant biomarkers, should be included to allow reproducibility.

Author’s reply:

We thank the reviewer for the suggestions. The information about HF and biomarker is in the previous polished cohort profile. However, to make sure we provide clarity for reproducibility. We include the following sentences in method section to clarify the HF classification “Heart failure with reduced ejection fraction (HFrEF) was defined as symptomatic heart failure with a left ventricular ejection fraction (LVEF) ≤40%; heart failure with mildly reduced ejection fraction (HFmrEF) as symptomatic heart failure with LVEF 41–49%; and heart failure with preserved ejection fraction (HFpEF) as symptomatic heart failure with LVEF ≥50%, consistent with the 2022 AHA/ACC/HFSA guideline definitions.” We also include the HIV biomarker classification “Controlled HIV is defined by HIV viral load less than 200 copies/mL.”

Statistical analyses need to be more explicitly described, including any adjustments for confounding variables such as age, sex, comorbidities, and antiretroviral therapy use; consideration of interaction terms or stratified analyses might also enhance interpretation.

Author reply:

1) The proportional hazard model was adjusted with age, sex, baseline EF, controlled HIV, and comorbidities such as chronic obstructive pulmonary disease (COPD), end-stage renal disease (ESRD), cancer, hyperlipidemia, hypertension, diabetes mellitus, peripheral artery disease (PAD), pulmonary hypertension, and coronary artery disease (CAD), ADL and smoking status. It was mentioned in the results section and now moved up to the method section.

2) Effect modification is now tested and data is included in supplementary table 2. Sensitivity analysis was done within patient with controlled HIV and HFrEF. The data of sensitivity analysis were included in the supplementary table 3.

The results section would benefit from more granular reporting, particularly regarding subgroup differences and effect sizes, rather than focusing solely on statistical significance.

Authors reply:

We appreciate reviewer’s comments on the description of characteristics across different racial groups. We have included more details in result section to address the subtle differences that may be overlook because of the sample size limitations.

Discussion of potential limitations is limited; the authors should address issues such as residual confounding, single-center design, and potential selection bias, and how these factors might influence generalizability.

Authors reply:

We thank reviewer for the suggestions. The add additional limitation of residual confounding and potential selection bias, and how it may influence the generalizability.

“Despite extensive covariate adjustment, residual confounding is an inherent limitation of observational cohort studies and results should be interpreted with caution. However, our sensitivity analyses yielded consistent findings, supporting the robustness of the results.”

“Furthermore, our cohort originates from New York City's largest safety-net hospital system, which is committed to providing healthcare services to the most vulnerable population, regardless of socioeconomic status, insurance coverage, or immigration status. This commitment ensures a unique study population that may significantly differ from those typically served by private healthcare systems, creating the selection bias and limited generalization.

Additionally, the discussion could more thoroughly integrate the findings with existing literature, highlighting similarities, differences, and potential mechanistic explanations.

Authors reply:

We appreciate reviewer’s suggestions for revising discussion. The discussion is now extensively re-written with more integrations with current literature.

Finally, attention to clarity in tables, figures, and text—ensuring consistency in terminology and proper labeling—would improve readability.

Authors reply:

We thank reviewer for the suggestions. We have review and edit the tables, figures, and text—ensuring consistency in terminology and proper labeling

Overall, the manuscript addresses a clinically relevant topic, but these areas of improvement would enhance rigor, clarity, and interpretability.

Journal Requirements:

Author reply: We thanks the reviewer for the instruction. It is now compliant with journal guideline

2. We note you have included a table to which you do not refer in the text of your manuscript. Please ensure that you refer to Table 3 in your text; if accepted, production will need this reference to link the reader to the Table.

Author reply: All figures/captions/table are all placed right after the paragraph in which they were first cited.

3. We note that there is identifying data in the Supporting Information file <file name>. Due to the inclusion of these potentially identifying data, we have removed this file from your file inventory. Prior to sharing human research participant data, authors should consult with an ethics committee to ensure data are shared in accordance with participant consent and all applicable local laws.

Author reply: All data are now removed from the portal. It will not be open to public. But the research committees will consider reasonable request.

4. Please include captions for your Supporting Information files at the end of your manuscript, and update any in-text citations to match accordingly. Please see our Supporting Information guidelines for more information: http://journals.plos.org/ploso

---

## [Decision Letter · Decision Letter 1]

14 Jan 2026

Dear Dr. Chen,

Thank you for submitting your manuscript to PLOS ONE. After careful consideration, we feel that it has merit but does not fully meet PLOS ONE’s publication criteria as it currently stands. Therefore, we invite you to submit a revised version of the manuscript that addresses the points raised during the review process.

We look forward to receiving your revised manuscript.

Kind regards,

Ronaldo Go, MD

Academic Editor

PLOS One

Journal Requirements:

Reviewer's Responses to Questions

**Comments to the Author**

Reviewer #2: All comments have been addressed

Reviewer #3: All comments have been addressed

2. Is the manuscript technically sound, and do the data support the conclusions?

Reviewer #2: Yes

Reviewer #3: Yes

3. Has the statistical analysis been performed appropriately and rigorously?

Reviewer #2: Yes

Reviewer #3: Yes

4. Have the authors made all data underlying the findings in their manuscript fully available?

Reviewer #2: Yes

Reviewer #3: Yes

5. Is the manuscript presented in an intelligible fashion and written in standard English?

Reviewer #2: Yes

Reviewer #3: Yes

Reviewer #2: Thank you for the thorough revisions. All previous concerns have been addressed, and the manuscript is now clear, rigorous, and suitable for publication, with only minor editorial polishing needed.

Reviewer #3: (No Response)

**Do you want your identity to be public for this peer review?** For information about this choice, including consent withdrawal, please see our Privacy Policy

Reviewer #2: **Yes:** Edgar Muchinta

Reviewer #3: **Yes:** Chukwuka Elendu

---

## [Author Response · Author response to Decision Letter 2]

16 Jan 2026

We are grateful to the reviewer for their careful and thoughtful review, which has enhanced the quality of the manuscript and prompted meaningful academic discussion.

---

## [Editor Report · Decision Letter 2]

11 Feb 2026

Racial Differences in People Living with HIV and Heart Failure: Insight from New York City Health and Hospitals HIV Heart Failure Cohort

PONE-D-25-22933R2

Dear Dr. Chen,

We’re pleased to inform you that your manuscript has been judged scientifically suitable for publication and will be formally accepted for publication once it meets all outstanding technical requirements.

Kind regards,

Ronaldo Go, MD

Academic Editor

PLOS One
---

## [Editor Report · Acceptance letter]

PONE-D-25-22933R2

PLOS One

Dear Dr. Chen,

I'm pleased to inform you that your manuscript has been deemed suitable for publication in PLOS One. Congratulations! Your manuscript is now being handed over to our production team.

Kind regards,

on behalf of

Dr. Ronaldo Go

Academic Editor

PLOS One